# Early Days of SACLA XFEL

**Tetsuya Ishikawa**

RIKEN SPring-8 Center, Sayo 679-5148, Hyogo, Japan; ishikawa@spring8.or.jp

**Abstract:** The SACLA (SPring-8 Angstrom compact laser) was designed to significantly downsize the SASE (self-amplified spontaneous emission) type XFEL (X-ray free-electron laser), in order to generate coherent light in the wavelength region of 0.1 nm by adopting an in-vacuum undulator that can shorten the magnetic field period length. In addition, a SASE XFEL facility with a total length of 700 m has become a reality by using a C-band RF accelerating tube that enables a high acceleration gradient. Although progress was initially slow, the small-scale, low-cost SACLA was smoothly constructed, and it became the second light source to lase in the 0.1 nm wavelength region, following the LCLS (linac coherent light source) in the United States. In this paper, we look back on the history leading up to SACLA. and describe the SCSS (SPring-8 compact SASE source) project as a preparatory stage and a part of the construction/commissioning of SACLA. Since March 2012, SACLA has been operating as a shared user facility. Just a few of the upgrade activities of the facility and advanced research conducted are introduced. Finally, we will discuss the future development of the SPring-8 site, which has co-located the third-generation synchrotron radiation facility SPring-8 and the X-ray free-electron laser facility SACLA.

**Keywords:** compact XFEL; SACLA

## 1. Introduction

The SACLA is a hard X-ray free-electron laser based on the SASE (self-amplified spontaneous emission) principle collocated with the 3rd generation synchrotron radiation facility SPring-8 on the same site (Figure 1) [1]. The construction project of SACLA started in 2006, and the first laser oscillation was observed in June 2011, being the second hard X-ray free-electron laser in the world after LCLS [2] at the Stanford Linear Accelerator Center (SLAC), USA. After commissioning and tuning for several months, SACLA started user operation in February 2012. Since then, SACLA has continued to operate as a shared facility with continuous improvement.

The feature we aimed for in the construction of SACLA, which was not found in other facilities, was the downsizing of the SASE hard X-ray free-electron laser. Combining the high-frequency (C-band) normal-conducting linac [3] with a short-magnetic-period in-vacuum undulator developed at SPring-8 and KEK [4], we have achieved unprecedented downsizing of the SASE XFEL.

This paper describes some aspects of SACLA as a vital element of the SPring-8 complex. Starting with a brief historical description of the development of accelerator-based light sources, we look back on the progress of SACLA's construction project in detail until its completion with the start of user operation. Then, we describe the selected topics of advancement of accelerators, optical systems, and experimental equipment promoted in parallel with user operation.

Finally, we will discuss the future development and deployment of accelerator-based light sources throughout the SPring-8 site.

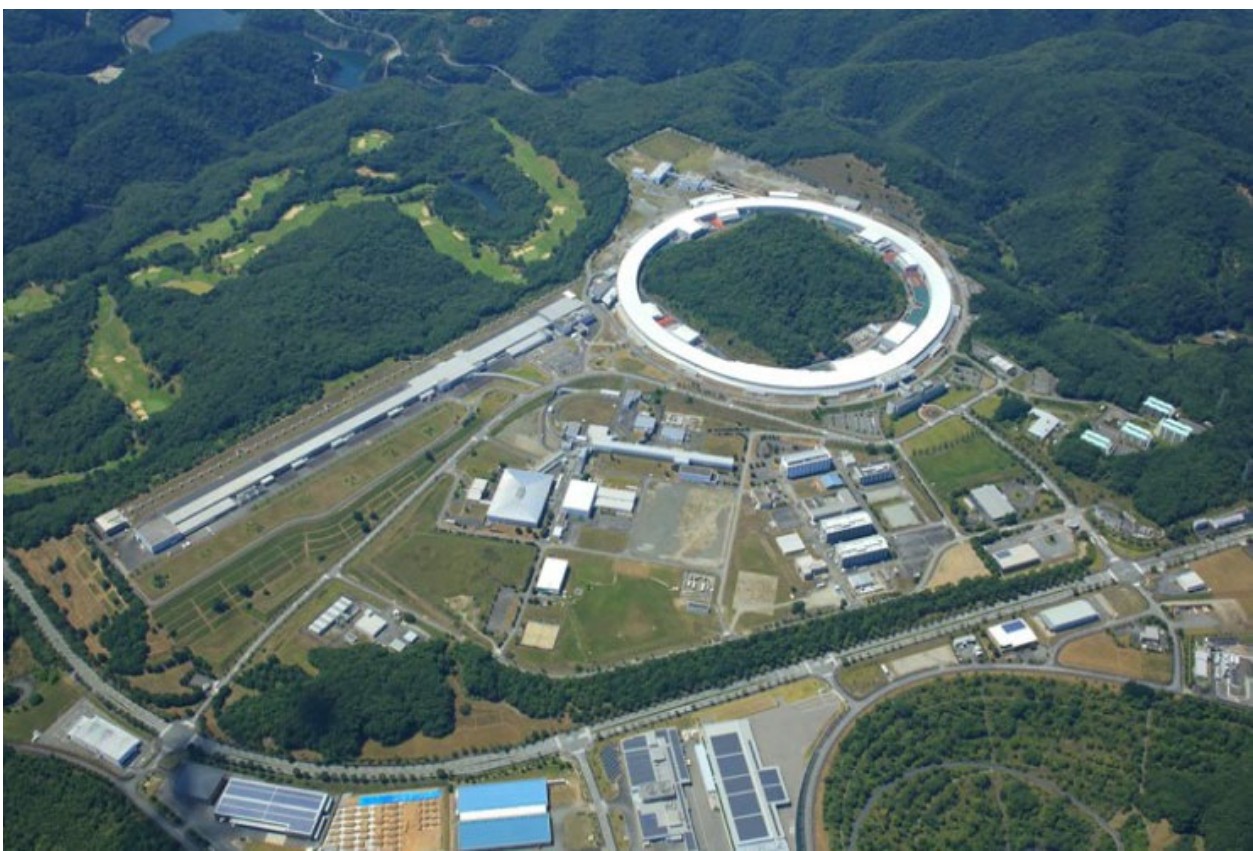

**Figure 1.** Aerial view of SPring-8 site. The circular building in the upper center is the SPring-8 storage ring building, and the long square building on the left is the SACLA XFEL building. The storage ring building is made by scraping the foot of a single rock to create a flat surface and placing the building on top. The rocky mountain remains unremoved in the central part.

## 2. The Road to SACLA

Recent advances in material science are supported by an understanding of matter's structure and electronic state. An electromagnetic probe is essential for obtaining such information experimentally [5]. Regarding to light sources on the longer wavelength side, such as visible and infrared light, the replacement of thermal sources with lasers is progressing. However, operating lasers for extreme ultraviolet (EUV) and shorter wavelengths was challenging. Therefore, accelerator-based synchrotron radiation has been used as a high-intensity light source in this region [6]. Synchrotron radiation is an electromagnetic wave emitted in the tangential direction of an orbit when a charged particle with a relativistic velocity changes its traveling direction. The utilization of synchrotron radiation initially began with the parasitic use of accelerators for elementary particle research. Still, as its usefulness was widely recognized, an accelerator dedicated to synchrotron radiation in the VUV to soft X-ray region was constructed in the 1970s. The parasitic use of the accelerators for elementary particle research is called the first-generation synchrotron radiation facility, and the use of dedicated light sources is called the second-generation synchrotron radiation facility.

In the 1980s, dedicated light sources in the X-ray region were to be built around the world. These are based on a 2–3 GeV storage ring and are designed to use synchrotron radiation from bending electromagnets that bend the orbit of the electrons. When the dedicated light source freed us from the restrictions of parasitism on elementary particle research and made it possible to optimize it as an accelerator light source, various fields of research and development progressed in that direction. In particular, efforts were made to improve the brightness by exploiting insertion devices, such as an undulator [7] on

the straight part of the ring accelerator. The undulator is also used for the free-electron laser described later, but it can generate high-intensity quasi-monochromatic light when inserted into the storage ring. In the 1990s, the development of storage ring light sources optimized for undulator utilization progressed. The light source itself does not have the coherence of a laser, but the spatial coherence is improved by looking at a small light source from a distance. The higher performance of the undulator requires reducing the amount of electron beam emittance, and low emittance rings using a Chasman–Green lattice [8] have become mainstream. A synchrotron radiation facility with a storage ring light source optimized for an undulator is called a third-generation synchrotron radiation facility. The Chasman–Green lattice has a magnet structure in which a deflection electromagnet is divided into two, and a convergence system is inserted between them. This is called a double-bend achromat (DBA). Undulator technology was immature in the 1980s when the third-generation synchrotron radiation facility was conceived. Therefore, there are many restrictions, and it was believed that high electron beam energy was required to emit hard X-rays with an undulator. The first hard X-ray third-generation light source was ESRF [9] in Europe, with an electron beam energy of 6 GeV. Subsequently, APS (7 GeV) [10] in the United States and SPring-8 (8 GeV) [11] in Japan were constructed (Figure 2). In addition to these, the German PETRA-III (6 GeV) [12], which converted the storage ring for elementary particle research into a synchrotron radiation facility, is also an example of a high-energy synchrotron radiation facility.

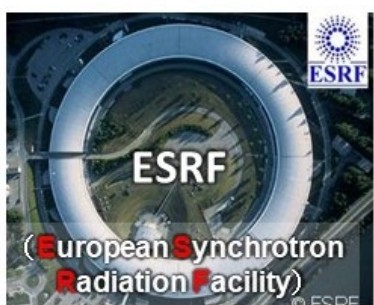 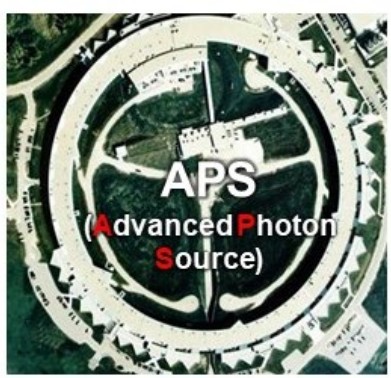 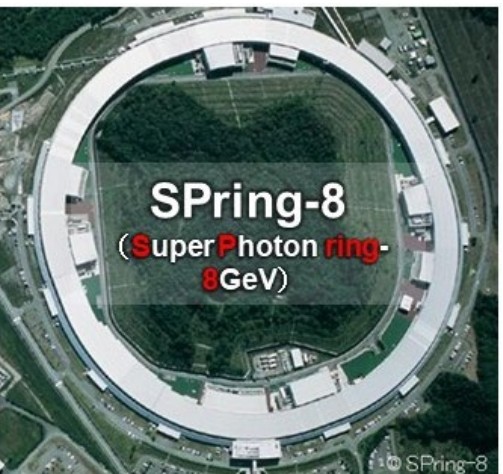

**Figure 2.** Early third-generation large synchrotron radiation facilities. Photos taken from directly above at the same scale. From the left, ESRF, APS, and SPring-8. The ESRF, which has been in operation since 1997, has an electron energy of 6 GeV, and a circumference of 844 m. The American APS, which has been in operation since 1996, has an electron energy of 7 GeV, and a circumference of 1104 m. Japan's SPring-8 started operation in 1997, but it has an electron beam energy of 8 GeV, and a circumference of 1436 m.

Third-generation synchrotron radiation facilities have made tremendous progress in undulator technology. The in-vacuum type [4] adopted by SPring-8 as a standard undulator makes it possible to generate hard X-rays even with electrons with lower energy. Therefore, the momentum for the construction of lower-cost medium-sized synchrotron radiation facilities (~3 GeV) has increased worldwide. For example, Switzerland [13] introduced SPring-8's in-vacuum undulator technology to construct a hard X-ray undulator beamline for protein crystallography. After that, in Europe, new 3 GeV synchrotron radiation facilities

were constructed in DIAMOND (UK) [14], SOLEIL (France) [15], and ALBA (Spain) [16]. Many second-generation synchrotron radiation facilities have also been refurbished to bring them closer to the third-generation standard. The NSLS-II [17] in the US and the TPS [18] in Taiwan have nearly become as advanced as possible in this direction.

Even while the actual deployment of third-generation synchrotron radiation facilities was in progress, theoretical consideration continued to keep an eye on future possibilities. One such possibility is the reduction of emittance using a multi-bend achromat lattice by Einfeld et al. [19]. The practical application of this idea began with MAX-IV [20] in Sweden, followed by SIRIUS [21] in Brazil and an ESRF upgrade [9]. Japan's Next Generation Synchrotron Radiation Facility in Sendai also adopted an MBA lattice. Construction projects of high-energy synchrotron radiation facilities with MBA lattices are underway in China [22] and South Korea. There are many plans to refurbish existing DBA-lattice-based facilities to MBA-lattice-based ones, and SPring-8 is also planning such a refurbishment [23]. In addition, renovation plans are underway at many existing facilities, such as APS-U [24], PETRA-IV [25], DIAMOND-II [26], and SLS 2.0 [27]. Synchrotron radiation facilities in the 2020s and 2030s are expected to move in this direction.

Let us return to the history, about 50 years before the present day. In 1971, John Madey proposed a free-electron laser (FEL) that generates coherent electromagnetic waves using free electrons as a laser medium [28]. Early FELs had a structure in which an undulator, that is, an optical emitter, was placed in an optical cavity. A feature of FEL lasers is that they have no principal limit to their wavelength. However, in reality, the reflectance of the normal-incidence mirrors constituting the optical cavity constrains the laser wavelength, making it almost impossible to perform a short wavelength FEL using an optical cavity. Nevertheless, FEL has been researched and developed worldwide as a new form of an accelerator-based light source. In Japan, FEL research and development has been promoted mainly in the infrared region at Osaka University [29], JAERI (currently QST) [30], Tokyo Science University [31], and Nihon University [32].

The discovery of the SASE principle in the 1980s [33] opened up the possibility of shortening the wavelength of FEL. When a low-emittance high-energy electron beam passes through a long undulator, the interaction between the undulator light and the electrons forms an electron density modulation with the intervals of the radiation wavelength. When such a density-modulated electron bunch passes through an undulator, it emits coherent light. This is the SASE principle. Using the SASE principle, instead of utilizing optical cavities, FELs can be constructed by combining linear accelerators and long undulators that can accelerate small emittance electron beams. As a result, the wavelength limitation of the free-electron laser is practically removed, and a coherent light source in the X-ray region can be conceived.

The Stanford Linear Accelerator Center (SLAC) first planned a short-wavelength coherent light source using the SASE method [34]. Although SLAC was one of the centers of high-energy accelerator research in the United States, the frontier of particle physics moved to higher energy, accelerator-based high-energy physics experiments in the United States, which were gradually consolidated into the Fermi National Laboratory. On the other hand, SLAC had an ultra-high-energy particle physics activity using cosmic rays, but accelerators there gradually became more and more popular in material science applications. The collider ring SPEAR, which was once used for the discovery of J-ψ particles, was parasitically used by SR users, converted to the second-generation synchrotron radiation ring SSRL, and then reduced in emittance to achieve performance close to that of the third-generation. As part of the expansion of the role of the laboratory, SLAC planned to construct an XFEL using its 2-mile normal-conducting linear accelerator that has been active in high-energy physics experiments for many years. The first symposium in this direction was held in 1994. This symposium promoted the study of XFEL in the United States, which later led to the LCLS project. However, the undulator technology was immature and had the same problem as the early large-scale synchrotron radiation facilities mentioned above. Therefore, it was believed that a high-energy electron beam of ~15 GeV was required to

emit X-rays near 0.1 nm, which determined the energy parameter of the electron linear accelerator to be 15 GeV.

In Europe, in the latter half of the 1990s, the study of the electron-positron linear collider construction project named the TESLA project began at Deutsche Electron Synchrotron (DESY), the German high-energy research center [35]. The collider uses superconducting linear accelerators, which achieve a much higher repetition rate than the normal-conducting linear accelerator in the US plan. However, as is the fate of high-energy physics research, a considerable investment is required for the purpose of discovering new particles with higher energy. As there are difficulties in convincing taxpayers to fund such specific research, scientists tried to gain a general understanding by installing XFEL with broader applications.

Since the TESLA project depended on the performance of the superconducting linear accelerator, a 1 GeV prototype was constructed as a test facility on the site of DESY, and a SASE-FEL in the VUV SX region was constructed using this prototype [36]. That FEL facility is now known as FLASH. It was the first practical FEL in this wavelength range and has since been used as a shared facility. The entire TESLA plan was not realized due to the domestic situation in Germany, but the XFEL part did became independently materialized as a shared facility of the EU. It is the current European XFEL [37]. Since the European XFEL was planned based on the undulator technology of the 1990s, similar to the LCLS, it required electron beam energy of 15 GeV or more to make an FEL with a wavelength of 0.1 nm or less.

Further, since the superconducting linear accelerator has a smaller acceleration gradient than the normal-conducting linear accelerator, the length of the accelerator for obtaining the same electron beam energy becomes longer. An electron gun will be placed on the DESY site, heading west-northwest through the deep underground of Hamburg, and an experimental facility will be set up in Schoenfeld, Schleswig-Holstein, 3.4 km away, where the user experiments will be conducted. The linear accelerator has a length of 2.1 km, and electrons are accelerated up to 17.5 GeV.

SPring-8 is a third-generation synchrotron radiation facility optimized for the use of insertion devices. Compared with 6 GeV of ESRF and 7 GeV of APS, the electron beam energy of SPring-8 is higher at 8 GeV, thus, making it easier to generate high-energy photons. However, at the same time, the heat load due to photons becomes higher, requiring many development factors, including the cooling of optical elements. When the discussion of SASE XFEL was in progress in the United States and Europe, SPring-8 was in the middle of commissioning in Japan. At that time, undulators and wigglers were discussed for the insertion devices. In both cases, a periodic alternating magnetic field is formed in the straight section of the storage ring by using many permanent magnet pieces, as shown in Figure 3. The device is called a 'wiggler' when the magnetic field is strong, for which the synchrotron radiation generated by the oscillatory electron orbit is superimposed incoherently. On the other hand, the radiation generated in the undulator superposes coherently to interfere constructively at a specific wavelength determined by the magnetic period length and the magnetic field strength, resulting in brighter quasi-monochromatic photons [5].

In SPring-8, we decided to use an insertion device configuration centered on the undulator and use the in-vacuum as the standard for the undulator [38]. An ultra-high magnetic field NdFeB magnet developed in Japan is used as a permanent magnet. Before developing the in-vacuum type, undulators had an ultra-high vacuum duct that allowed electrons to pass between the magnetic poles, limiting the gap width between the magnetic poles. Therefore, if the magnetic period length, $\lambda_m$, is too small, a magnetic field cannot be generated at the electron's position, imposing a minimum value of $\lambda_m$. The in-vacuum undulator has a structure in which the magnet arrays are placed in an ultra-high vacuum chamber. NdFeB magnets are porous materials which are very incompatible with the ultra-high vacuum. However, the Kitamura group of SPring-8 has developed an ultra-high vacuum compatible NdFeB magnet by coating it with a thin film of TiN [39]. Since there is no vacuum chamber between the magnet arrays in the in-vacuum undulator, the gap

between the magnet arrays can be reduced to the limit where it does not interfere with the electron beam. Then, even if $\lambda_m$ is small, it is possible to create a magnetic field at the electron beam position. As a result, the wavelength of the undulator fundamental radiation, $\lambda_p$, given by the following,

$$\lambda_p = \frac{\lambda_m}{2\gamma^2}\left[1 + \frac{k^2}{2} + \gamma^2\theta^2\right] \qquad (1)$$

becomes shorter with a small $\lambda_m$ if the electron energy is the same. Here, $\gamma$ is the Lorentz factor and $\theta$ the offet angle from the optical axis. The undulator parameter, $K$, is a function of the maximum magnetic field $B_0$ of the undulator and given by the following,

$$K = \frac{eB_0\lambda_m}{2\pi mc} \qquad (2)$$

where $e$ is the elementary charge, $m$ the electron rest mass, and $c$ the speed of light in vacuum. The equivalent $\lambda_p$ is obtainable with lower electron energy, $\gamma$, by reducing the value of $\lambda_m$. In addition, one of the developments of the undulator technology at the third-generation synchrotron radiation facility is the magnetic field adjustment technology, and the uniformity of the magnetic field is improved, resulting in the intensity enhancement of the higher harmonics. From these facts, as mentioned earlier, it became possible to cover a wide range of photon energy with medium-energy third-generation synchrotron radiation facilities, and the world moved in the direction of higher cost-performance [40].

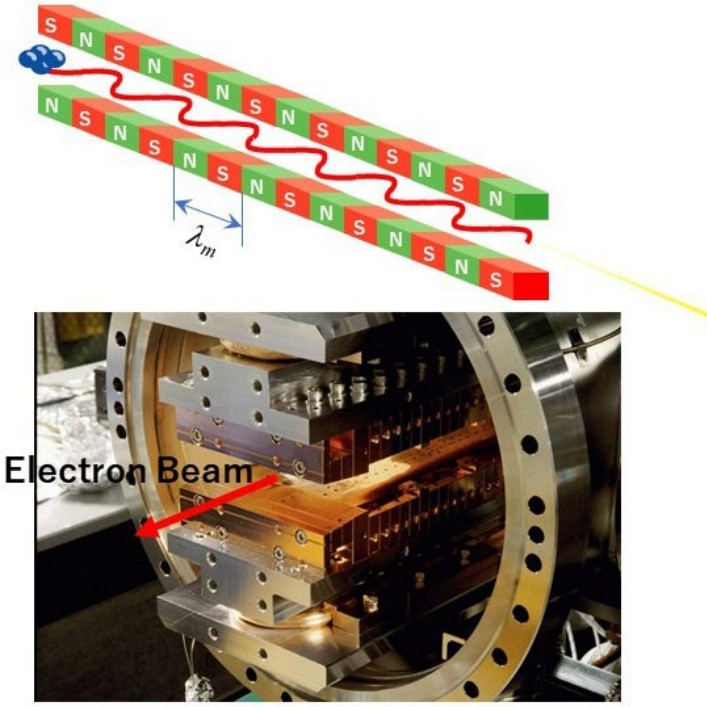

**Figure 3. Top**: Schematic diagram of the insertion device. In many cases, permanent magnets form a periodic magnetic field in the linear part of the accelerator, through which a relativistic electron beam is passed to generate synchrotron radiation. **Bottom**: Inside the SPring-8 standard in-vacuum undulator.

Compared to other third-generation high-energy synchrotron radiation facilities, such as ESRF and APS, SPring-8 has a larger storage ring diameter and a longer circumference due to the higher energy. In addition, the storage ring has four long straight sections, which makes it possible to install a long undulator. The width of the experimental hole is 20 m, which is the same as ESRF, but the length of the synchrotron radiation beamline extending in the tangential direction could be increased, thanks to the large diameter. The length of the

most extended beamline inside the experimental hall, without interfering with the passage provided at the outermost periphery, is 90 m. In addition to this, a space was provided where a long beamline protruding outside the storage ring building could be installed, making it possible to construct five 300 m beamlines and three 1000 m beamlines [41].

Looking at the low emittance electron beam of third-generation synchrotron radiation at a distance from the light source improves spatial coherence due to propagation, enabling coherence-related measurements not possible with earlier-generation synchrotron radiation facilities. In particular, when the 1000 m beamline was constructed in the year 2000, and the monochromatic X-ray beam was first introduced to the end station, the first observation was the speckle of the beryllium window [42]. This observation strongly impressed the importance of coherence.

Prof. Yamauchi's group from Osaka University used this beamline to develop ultra-precision total reflection mirrors. The coherent incident light clarified various properties of the mirror and significantly contributed to improving the manufacturing method [43]. The mirrors developed here are named OSAKA MIRROR and are used in synchrotron radiation facilities and XFEL facilities worldwide. In 2001, the 27 m long undulator beamline was completed, and research and development were promoted on various problems that occur when the undulator is lengthened [44]. Since R & D at SPring-8 covered a part of the significant R & D elements at XFEL, Dr. Hideo Kitamura, who was in charge of undulator development, and the present author, who was in charge of optics development, were often invited to participate in the LCLS-related preparatory meeting and the TESLA XFEL-related preparatory meeting. Returning from these meetings, we discussed that the short-period in-vacuum undulator developed at SPring-8 could be used to realize a hard X-ray free-electron laser with a much smaller size. Therefore, targeting RIKEN's internal R & D funds in 2000, we submitted a proposal to develop various elemental technologies in the 5 years from 2001 to 2005 and to construct an SX-FEL with a 1 GeV linear accelerator as a prototype. The proposal was approved, and a research and development project was started in 2001.

As an aside, one of the meetings at DESY invited Jianwei Miao, the current UCLA professor (then a SLAC staff scientist) who was attracting attention to the development of coherent diffraction imaging [45]. He stopped by during a coffee break and asked if SPring-8 could do a hard X-ray CDI. It seems that he did not get on well in the United States. After the immediate OK, we decided to start joint research on the development of hard X-ray CDI. Hard X-ray CDI was established at SPring-8, and young people participating in the experiment as postdocs and graduate students continue to study CDI at various facilities and universities. Tamasaku developed nonlinear X-ray optical research with a 27 m undulator beamline to utilize X-ray coherence from another direction [46]. It has developed into a research program at SACLA and has produced different results.

## 3. SPring-8 Compact SASE Source (SCSS) Project

In 2001, we started an elemental technology development project with the immediate goal of a SX SASE light source using a 1 GeV linear accelerator. At first, we planned to use S-band accelerating tubes for the linac. However, KEK was developing C-band and X-band accelerating tubes with higher acceleration gradients for the linear collider, so we considered using them. As a result, we decided to adopt a C-band accelerating tube, which was considered to be more feasible at that time, and we asked for the help of Professor Shintake, who was in charge of the development of the C-band tube at KEK [47]. By combining a C-band accelerating tube with a high acceleration energy gradient and a short magnetic field period undulator, the aim was to further reduce the size of the SASE type free-electron laser that emits high-energy light. In the end, Professor Shintake came to RIKEN, and we asked him to supervise the accelerator development of the elemental technology development project. We asked Dr. Kitamura to supervise the undulator development. The present author supervised the entire project and the development of X-ray optics.

The development of the C-band acceleration tube and the klystron for the C-band has been completed considerably at KEK, but the configuration of SASE FEL demanded the completion of the electron gun and the injection part. Since SASE FEL requires a short bunch electron beam, it is common to cut out a short bunch e-beam from the beginning using a laser RF electron gun, accelerate it, and further compress it in the longitudinal direction. The laser RF gun is also used in LCLS and FLASH. We decided not to adopt the laser RF gun, but instead to use the conventional thermionic gun. The $CeB_6$ single crystal used for the cathode of the electron microscope was adopted for the cathode of the gun. The laser RF electron gun draws a large charge with high density, so we were worried that the Coulomb repulsion between the electrons would degrade the emittance. We decided to develop a method to draw out a low-density electron flow with a thermionic gun, cut it out in 1 ns, and gradually compress it in the time direction. We also worked on shortening the period of the in-vacuum undulator. We have developed an undulator with a magnetic field period of 15 mm, which was less than half of that for the standard undulators in SPring-8, which is 32 mm. The development of acceleration tubes and electron guns was completed around 2003, and the development of short-period undulators was also completed around the same time. In addition, the conceptual design of velocity bunching, which combines RF cavities with different frequencies and compresses electron bunches in the time direction while accelerating the electron bunches, has been completed. Therefore, along with starting the design of the prototype machine, the accelerator radiation shield tunnel was constructed by remodeling the existing building. Due to budgetary constraints, the energy of the prototype linear accelerator was reduced from its initial plan of 1 GeV to 250 MeV [48,49].

In parallel with the construction of the prototype, the design of the Å laser began. We took the policy of making an actual machine as an expanded version of the prototype machine. We constructed an undulator with a magnetic period of 15 mm for the prototype machine. Still, we found that it was better to slightly relax the conditions in order to improve the reliability of daily operation, so we decided to adopt a magnetic field period of 18 mm. We assumed 6 GeV as the energy of the linear accelerator to reach a wavelength of 0.1 nm with a magnetic period of 18 mm. In addition, we decided to place a distribution magnet on the downstream side of the linear accelerator so that the electron beam could be transferred to the five FEL lines. The installation location is assumed to be the land on the embankment ground beside the 1 km beamline.

The full-energy injection to SPring-8 was discussed during the study, changing the linac energy from 6 to 8 GeV. Since the magnetic period of the undulator was not changed, the shortest wavelength was 0.06 nm or less as a result. The results of this study were published as the Conceptual Design Report. While continuing such examination work, the "Program for National Critical Technology" was launched as one of the national science and technology promotion programs, and it selected candidates for promotion. The examination process decided to construct an X-ray free-electron laser as one of the national critical technologies.

After completing the conceptual design, an international advisory committee meeting was convened in October 2004, but the occurrence of a large typhoon prevented it from taking place as planned. We postponed the meeting to February 2005. The members invited were Prof. Namkung of POSTEC, Prof. Kurokawa of KEK, Prof. Couprie of LURE, Prof Galayda of SLAC, Prof. Schneider of DESY, Prof. Hastings of SLAC, and Dr. Kim of Argonne. The summary of the review committee was as follows [50]:

- SPring-8 compact SASE source (SCSS) is an innovative project for the generation and use of an intense, coherent, short pulse X-ray beam.
- The SCSS is unique due to its compactness and its co-location with SPring-8, the leading third-generation X-ray facility.
- The SPring-8 site and facility are very well-suited for this project.
- The success of the SCSS will be a milestone event in advance of technology for X-ray FEL, stimulating progress in X-ray science worldwide.

- Members of the SCSS project team are well-known for their success in innovative solutions to different problems.
- The project schedule of the SCSS is ambitious but feasible in view of the competence of the project team and accumulated knowledge in the SPring-8 site.
- The 60 nm FEL using 250 MeV accelerator is a major step toward the success of the project. We strongly recommend the provision for full scientific utilization of their unique source.
- The committee strongly recommends the prompt start of the SCSS project.

On 12 April 2005, the Japanese Society for Synchrotron Radiation convened the Special Committee Meeting for the Next Generation Light Source. Here, SPring-8's X-ray free-electron laser, KEK-PF's energy recovery linac light source, and the ultra-low emitter 3 GeV storage ring light source of the Institute for Solid State Physics, University of Tokyo, were discussed as candidates for national core technology. The energy recovery linac light source was discussed as a rival to SLAC LCLS in the United States. It uses a superconducting linear accelerator to accelerate an ultra-low emission electron beam, passing through an undulator. Since the undulator emits incoherent light, it is weaker than the coherent emission of XFEL if the number of related electrons is similar. However, it gains intensity by increasing the number of electrons. After generating X-rays, electrons are decelerated by the same superconducting linear accelerator that has accelerated them, and the energy is recovered and used for accelerating the next electron bunch.

When the number of electrons increases, the energy required for acceleration becomes very high, but power consumption is compensated by recovering energy during deceleration after the electrons emit light. However, ERL was not technically established at that time. Moreover, there is no doubt that, depending on the energy recovery rate, even recovery will result in a large amount of power consumption. Therefore, it was concluded that ERL is not suitable as a candidate for the national critical technology, but it was recommended to continue research and development. The ultra-low emittance 3 GeV storage ring light source focused on soft X-rays and VUV and excluded hard X-rays, so not all the synchrotron radiation researchers supported it. If this were a proposal to cover even hard X-rays, it might have become a candidate for a 3 GeV medium-sized storage ring light source later, but since it was proposed as a facility dedicated to low-energy photons, it was not supported by all academic societies. As a result, it was decided to construct a SASE light source at the SPring-8 site as a national critical technology. At the same time, next-generation supercomputer development, an ocean earth observation system, space development, and fast breeder reactor development were selected as national core technologies.

## 4. SACLA Construction and Commissioning

After being nominated as a candidate for the national critical technology, the facility's detailed engineering design study commenced. The project's primary purpose is to complete a compact SASE source for a 0.1 nm wavelength based on the new concept described above. Additional functions (i) as an injector to SPring-8 and (ii) the simultaneous SPring-8 and XFEL utilization were also discussed. We plan to fully utilize the advantages of the co-location of the third-generation synchrotron radiation facility and the X-ray free-electron laser.

The project construction period was 5 years, from 2006 to 2010. The whole construction was directed by Dr. Noritaka Kumagai, who had directed the accelerator construction of SPring-8. The XFEL was supposed to start the shared operation by the end of FY2011, after a little less than a year of start-up adjustment operation. The construction budget was initially estimated as 50 billion JPY but was eventually allocated as just under 40 billion JPY, the difference being absorbed by doing all engineering works in-house. Details of the overall layout were examined, including the ground conditions. Electromagnets can correct the trajectory of the electron beam, but the light travels straight. Causing SASE requires light and electrons to travel together in an undulator over 100 m with an accuracy of several micrometers. This condition urged us to install the undulator in a very stable building. The

stability required for the accelerator is lower than that of the undulator. Many experiments using X-ray focusing at several nanometers also require an extremely stable building. Based on these considerations, we put the start of the linear accelerator near the 1 km experimental station, accelerating the electrons toward the SPring-8 storage ring building (Figure 4). The base of the linear accelerator building was formed by many piles reaching the bedrock about 50 m underground, on which a long and narrow building was built. For the undulator building and the utilization research building, where stability is essential, the artificial rock created to connect to the bedrock formed the base of the building. The utilization research building has an experiment hall, research offices, experiment preparation rooms, an auditorium, and a computer server room.

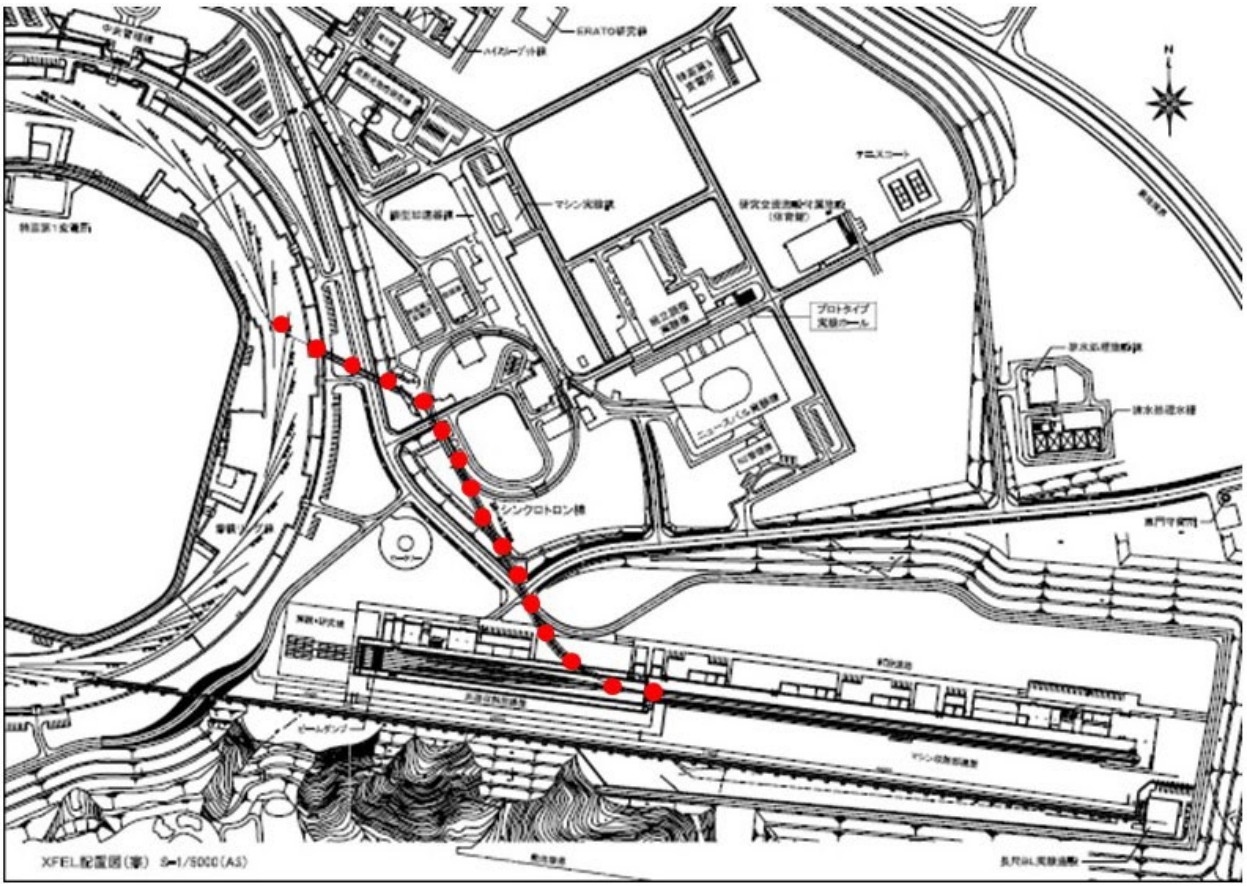

**Figure 4.** Plan view map of a part of the SPring-8 site. The straight building at the bottom of the map is the SACLA facility. Electrons generated by a gun located at the rightmost part of the facility are accelerated to the left and head toward the SPring-8 direction. Red dots show the electron beam transport from the SACLA linear accelerator to the SPring-8 storage ring.

The outline of the project schedule is shown in Figure 5. The contract work for the accelerator building design started at the beginning of FY2006 and was completed in February 2007. The contract work for the construction of the accelerator building began in January 2007, followed by the construction work between March FY2007 and December FY2008. The undulator building started construction in 2007 and was finished at the end of 2008. The construction of the experimental building began at the end of 2008 and was completed in 2010. The injector's engineering design, including the electron gun, was finished in October 2006. All items were delivered at the end of FY2007. After the offline examinations in 2008, installation and adjustments in the accelerator building began in 2009. Detailed design of some accelerator components with shorter delivery time was completed in the first half of 2006, followed by a procurement process. All components delivered by the end of 2008 were used for the installation work at the accelerator building, which

was started in 2009. The remaining components were delivered in FY2010 and installed at the end of 2010. Undulator and beamline equipment started contract work in 2007 and were to be delivered by FY2010. In addition to the above, an electron beam transport to the SPring-8 Booster Synchrotron was constructed in 2010.

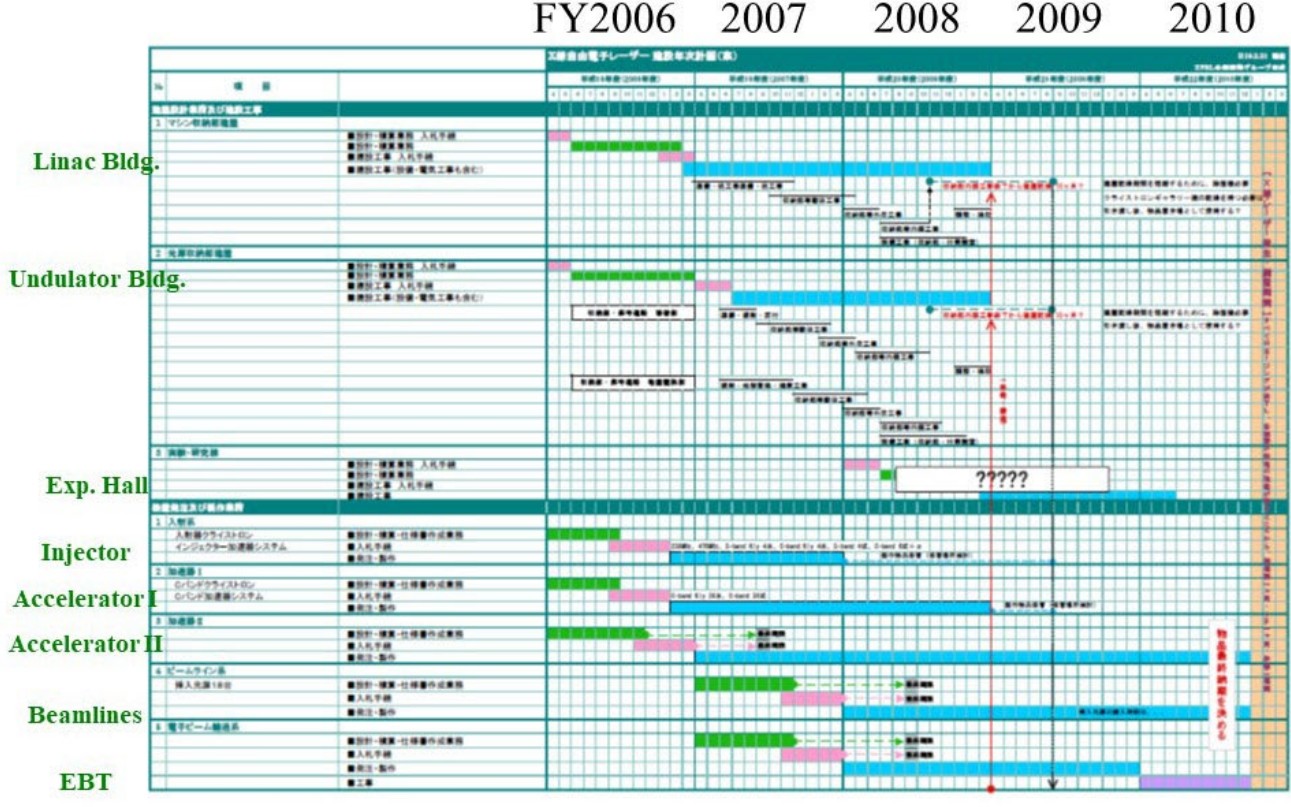

**Figure 5.** SACLA construction project schedule for 2006–2010. Downsizing has made it possible to build facilities in five years.

The undulator building and experimental hall accommodated up to five undulator lines. The characteristics of the electron beam forced us to change the distribution of the five lines to be parallel to each other instead of in the original fan shape. The first XFEL line was installed on BL3, extending straight from the linear accelerator, and the outermost BL1 was set as a line using wideband short pulse synchrotron radiation from VUV to SX. The safety interlock system was built based on the same concept as SPring-8. However, the difference is that linear accelerators discard electrons in the beam dump every time. A permanent magnet reflector was installed downstream of the electromagnet to prevent electrons from reaching the downstream laboratory in a beam dump electromagnet failure. Elections that accidentally traveled straight were bent here and did not penetrate downstream. In June 2006, the prototype machine oscillated at a wavelength of 49 nm, where test development related to the 8 GeV accelerator was carried out, and various technological developments, such as measurement systems and timing systems for use were carried out. The prototype was later relocated to BL1 in the undulator building and used for the VUV-SX FEL in BL1.

Details of the facility are described in the literature [1]. The design parameters are shown in Table 1. Construction was proceeding smoothly, but here we move on to the topics of the promotion of utilization.

**Table 1.** Main parameter list for SACLA.

| Parameter | Present Value |
| --- | --- |
| Electron beam | |
| Beam energy | 8.5 GeV (maximum) |
| Beam charge | ~0.2–0.3 nC |
| Peak current | >3 kA |
| Energy spread (projected) | <0.1% |
| Normalized emittance (projected) | $1\pi$ mm mrad |
| Repetition rate | 60 Hz |
| Undulator | |
| Periodic length | 18 mm |
| Number of undulator modules | 18 |
| Total number of periods | 4986 |
| Maximum K | 2.2 |
| Minimum gap | 3.5 mm |

When SACLA started, there was no operating XFEL in the world. However, at the existing LCLS in the United States and the DESY in Germany, discussions were underway on new science that would be feasible for the first time at XFEL. The SACLA in Japan, LCLS in the US, and TESLA-XFEL in German collaboratively held a 3-site meeting on XFEL in 2007. The purpose was to promote the XFEL science and technology by cooperating to solve common problems of accelerator technology and utilization technology, and to exchange information and human resources. The second conference was held at SLAC in 2009, and the third conference was held at DESY in 2010. Later, Swiss XFEL and PAL XFEL in South Korea joined, resulting in a 5-site meeting. In 2019, LCLS hosted the 10th meeting. The European XFEL was supposed to preside over the 11th meeting in 2020, but COVID-19 postponed it. Despite the impact of the coronavirus, cooperation continues as a series of online meetings.

In Japan, the momentum for launching new science for XFEL has increased, and MEXT has created a program for that purpose as the utilization promotion research subject. Under the MEXT, the Utilization Promotion Research Council was established with the Utilization Promotion Policy Formulation Project Team and the Utilization Promotion Research Project Selection/Evaluation Project Team. A public call gathered research teams from 25 universities, and research institutes in Japan and overseas, to cooperate with RIKEN to prepare the utilization of an 8 GeV machine and a 250 MeV prototype machine. The program was implemented from 2006 to 2010 in parallel with the facility's construction. The first three years were the elemental technology development period, and the following two years were the integrated system development period. A total of 18 subjects proceeded in parallel. The subjects were broadly divided into elementary technology development, material science-related development, and life science-related development.

The SACLA hardware was completed by the end of the fiscal year, although the Great East Japan Earthquake at the end of the 2010FY caused some delays in delivering minor components. Construction of SACLA went smoothly, and the accelerator conditioning operation began in the fourth quarter of the 2010FY. At the end of the 2010FY, we achieved the 8 GeV operation of the linear accelerator and observed the spontaneous undulator radiation. From the beginning of the 2011FY, the commissioning was started. The adjustment of the accelerator went smoothly and, finally, the adjustment of the undulator, which determines the success or failure of the XFEL, began.

From the beginning of the construction project, major unresolved issues for XFEL development were the detectors and the alignment of undulators.

XFEL emits a tremendous number of photons in a femtosecond pulse. Since the sample often breaks when one energetic pulse hits the sample, replacing the sample for each pulse is necessary. In addition, it is necessary to analyze experimental data considering fluctuations in pulse characteristics of SASE. The preferable solution is to develop an X-ray two-dimensional detector system that can acquire experimental data synchronizing with the XFEL pulse. Unavailability of such detectors when the XFEL development started urged LCLS and Euro XFEL to set up detector development programs. SACLA has decided to select the multi-port CCD technology for its detector, as this can be reliably used at the start of service. This detector is capable of high-speed, low-noise readout corresponding to the SACLA repetition rate of 60 Hz. In the XFEL experiment, various experimental schemes have been proposed, but the required actual performance of the X-ray two-dimensional detector is similar. Therefore, SACLA decided to develop a detector platform that can flexibly handle various experiments instead of optimizing multiple detectors for each experiment. Each component, including the sensor, is modularized in this platform. The difference in module configuration absorbs the difference between experiments [51].

For example, beam diagnostic spectroscopes and standard Bragg diffraction measurements use small detectors equipped with only one MPCCD sensor, whereas coherent diffraction microscopes and protein microcrystal structure analysis use more giant detectors with eight sensors arranged in a mosaic pattern (MPCCD Octal Sensor Detector). Table 2 shows the performance of the MPCCD sensor. A readout speed of 60 frames/s is achieved by reading in parallel from the 8 readout ports of the sensor. Massive data from MPCCD is processed by the DAQ system developed for SACLA, which is equipped with a supercomputer for data analysis [52].

**Table 2.** Main parameter list for MPCCD.

| Description | Parameter | Unit |
| --- | --- | --- |
| Pixel size | $50 \times 50$ | μm |
| Pixel Number | $1024 \times 512$ | N/A |
| Image area | $51.2 \times 25.6$ | mm$^2$ |
| Sensing Material | Epitaxial Silicon | N/A |
| Sensng Layer Thickness | 50 | μm |
| Sensor structure | Front-illumination | N/A |
| Image format | Full frame transfer | N/A |
| Operation temperature | 0 to 30 | C |
| Quantum efficiency | 80 | % at 6 keV |
| | 20 | % at 12 keV |
| Max. frame rate in un-binned mode | 60 | Hz |
| Max. pixel readout speed | 5.4 | MHz |
| Readout port | 8 | N/A |

X-ray free-electron lasers (XFELs) with long undulators made up of many segments have many sources of error to reduce FEL gains, such as trajectory errors, K-value mismatches, and phase mismatches. These are related to the segmented undulator structure. Undulator commissioning refers to the tuning process to eliminate possible error sources and is an essential step towards achieving lasing. At the SPring-8 Angstrom compact free electron laser (SACLA) facility, undulator commissioning is performed by characterizing X-ray radiation, namely spatial and spectral profile measurements of monochromatic and spontaneous undulator radiation, and FEL intensity measurements.

The undulator at SACLA consisted of 18 segments at initial commissioning, one of which was 5 m long when the first adjustments were made. The photon beam-based alignment [53], first performed at SACLA, has a variable slit 80 m downstream from the undulator outlet, and a Photon BPM [54] for measuring FEL radiation pulse energy downstream of the variable slit. After monochromatization with a double-crystal monochromator downstream, photon flux measurement is performed with a photodiode detector, or two-

dimensional X-ray spatial profile detection is performed with MPCCD. This alignment method is smart enough to accomplish the first lasing at 0.12 nm wavelength in a single month, on 7 June 2011. After observing the first lasing, we adjusted the accelerator and undulators further. Although not fully saturated, we reached the lasing at 0.063 nm in October. We reached the lasing at 0.225 nm in November for the longer wavelength side. Not surprisingly, from the current point of view, we have shown that XFEL can be reached over a wide wavelength range by changing the electron energy and adjusting the K-value. In particular, the fact that the laser wavelength can be changed by simply changing the K-value due to the change in the undulator gap dramatically reduces the time and effort required to change the wavelength and makes it possible to make changes in a manner that responds immediately to the user's request.

The beam position monitor (BPM) and double-crystal monochromator (DCM) were installed and adjusted, which was very useful for photon-beam-based undulator alignment. The initial photon beamline (BL3) was constructed parallel with the accelerator. Due to space constraints, the details of BL3 would be left to the report by Tono et al. [55].

With these preparations, SACLA started user operation in March 2012 as the second XFEL generating a 10 keV FEL.

## 5. Upgrade Activities after Inauguration

The BL3 was first used as SACLA's only XFEL beamline in various fields, from life sciences and materials sciences to extreme state sciences. Various trials have been made but relocating the equipment in each experiment reduced efficiency. Therefore, in 2013, we started the construction of the second FEL line, BL2. We started fine-tuning BL2 in October 2014 and released it to general users in March 2015. A photograph of the undulator hall is shown in Figure 6, in which two rows of long undulators are seen.

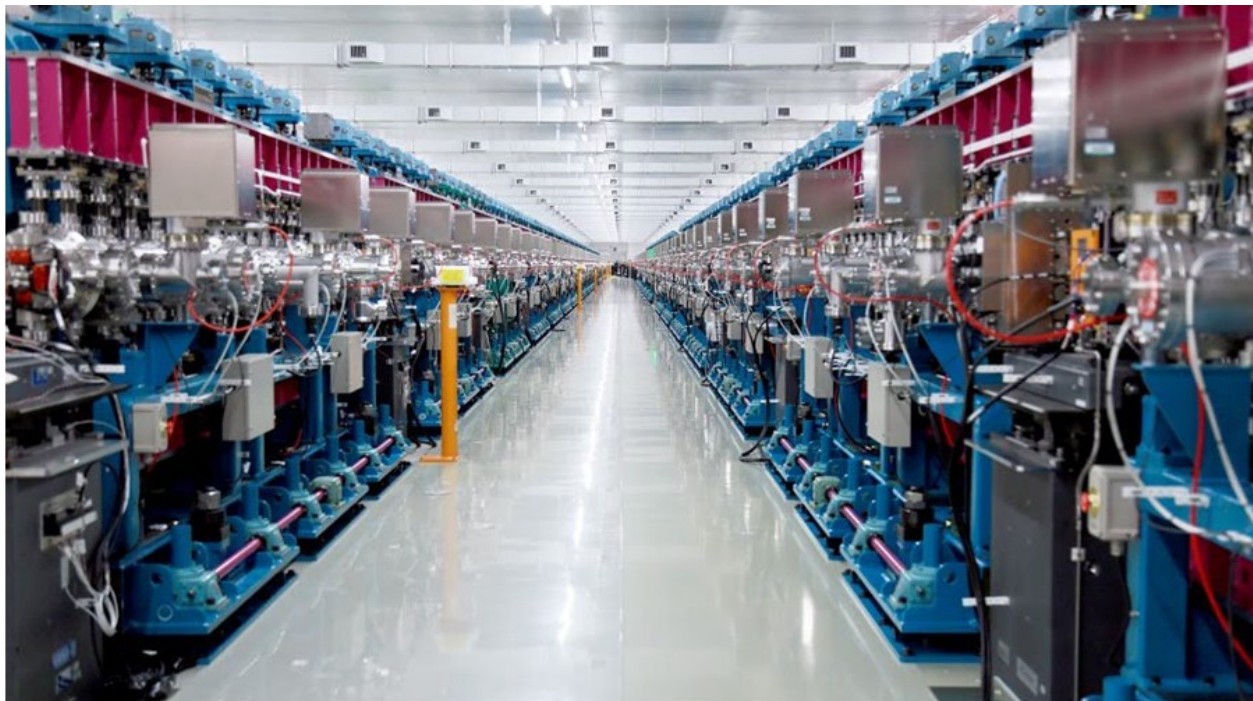

**Figure 6.** SACLA undulator hall: Two rows of ~100-m in-vacuum undulators producing XFEL. Left: BL3, right: BL2.

A time-interleaved multi-energy acceleration concept [56] was developed to simultaneously use BL2 and BL3 as XFELs with different energies. It is a method of accelerating the electron beam bunch coming at 60 Hz to different energies alternately, and the goal is to lase the two FEL lines stably with different energies. Acceleration energy is controlled by

modulating the RF phase of the electron accelerating tube. For example, if phase modulation is applied at 30 Hz to 60 Hz electron bunches, the adjacent bunches could be placed on the maximum acceleration and deceleration phases. Thus, electron bunches with different energies can be obtained alternately. These are distributed to BL2 and BL3 by a switching magnet. Then, electron bunches with different constant energies will pass through each undulator pulse-by-pulse [57].

One of the characteristic features of SACLA is to adopt the variable gap in-vacuum undulators, which offer a variety of new possibilities and the ability to change the XFEL wavelength easily. One is a two-color double-pulse (TCDP) FEL [58]. The ninth segment of the BL3 undulator row was moved to the most downstream part, and a chicane was installed to delay the electron beam. When this experiment was performed, the total number of undulator segments increased by one to 19. Setting the K-value of the upstream eight undulator segments to be $K_1$, one electron bunch generates an XFEL pulse of the corresponding wavelength. Then the electron bunch passes through the chicane and enters downstream undulator segments with different K-values, $K_2$, from $K_1$. The electron bunch generates an XFEL pulse with a different wavelength from the first XFEL pulse with a time delay determined by the chicane parameter. As a result, a two-color double-pulse (TCDP) is generated by changing the K-values of the undulator segments, with a wavelength difference of 30% and a time difference tunable with an accuracy in the attosecond region.

Collaboration with Osaka University in developing ultra-precision X-ray mirrors at SPring-8 [43] continues at SACLA. The ultra-precision mirrors developed at SPring-8 have come to be used in synchrotron radiation facilities worldwide as the OSAKA Mirror. In SPring-8, Kirkpatrick–Baez (KB) mirrors composed of two elliptical cylinder mirrors have generated 100 nm focused beam routinely. When considering mirror focusing with SACLA, there was concern about surface damage due to high-intensity XFEL. The surface should be composed of light elements, and the reflectivity should be close to 100% to have high radiation resistivity. It was necessary to reduce the glancing angle, but a smaller glancing angle indispensably lengthened the mirror to prevent the beam from being missed. Therefore, a technology for polishing a 400 mm long ultra-high precision mirror was developed. At the time of the XFEL lased in 2011, a KB arrangement of two 400 mm mirrors that focused 10 keV XFEL to a 1 μm spot was installed in BL3 [59]. The power density at the focal spot was $6 \times 10^{17}$ W/cm$^2$. In addition, a focal size of 50 nm was achieved by 2-stage focusing with 2 sets of KB mirrors [60]. The power density at the 50 nm focal spot was as high as $10^{20}$ W/cm$^2$. The 50 nm beam observed nonlinear X-ray processes, such as supersaturated absorption [61], proving exceptionally high-power density.

When designing an XFEL facility plan, the coherent diffraction imaging of a single protein molecule was a dream goal. However, the pulse energy of the SASE XFEL was not sufficient to give a high-resolution single-molecule image. In LCLS, an X-ray diffraction method using microcrystals was attempted, and applications developed rapidly. This method was named as serial femtosecond crystallography (SFX). In this method, many microcrystals are put into the beam one after another to acquire a diffraction pattern [62,63]. The crystal structure is determined by analyzing the diffraction patterns in random orientations. The XFEL pulses hitting the microcrystal samples break the samples, so an approach called diffraction before destruction is adopted, in which the samples are sent out one after another. Since SACLA has a very narrow pulse width of 10 fs or less, it is possible to collect diffraction data before a Coulomb explosion destroys the molecule [64].

The protein molecular structure analysis at SACLA uses two methods. One of these is for collecting high-resolution data using a goniometer and a high-quality crystal sample, similar to the synchrotron radiation experiment. The position of the sample is changed each time after the XFEL pulse irradiation, and the next pulse impinges on the new position of the crystal [65]. On the contrary, in SFX, minute crystals are dispersed in a liquid and continuously sent to the irradiation field to acquire many diffraction data.

Subsequently, DAPHNIS (a diverse application platform for hard X-ray diffraction in SACLA) was developed as a continuous X-ray diffraction experiment platform [66]. It

has been used for many protein SFX and crystallography measurements since it started to experience shared use in 2013. The DAPHNIS process consists of a sample chamber, a sample injector, and an X-ray image detector to enable 1 μm focused XFEL measurements from the beamline focusing mirror [59]. The image detector records the diffraction patterns from SACLA XFEL pulses repeating at a maximum of 60 Hz.

Here, we introduced the relatively early situation of the SACLA facility and did not touch on seeding development [67], split and delay measurement system development [68], optics for high-speed phenomenon research [69], imaging research [70], high-energy density research [71], and others. Please refer to the SACLA website [72] for the status of newer research.

## 6. Future Outlook

Around 2004, when we were conducting the construction campaign for SACLA, we made the first version of Figure 7. This showed that the SPring-8 site could succeed alongside the synchrotron radiation facility SPring-8 and the compact XFEL facility SACLA. The upgrade of SPring-8 (SPring-8-II) was assumed as the next step. After completing SACLA, the conceptual design of SPring-8-II began, which was planned to adopt an MBA lattice and to aim for a horizontal emittance of 100 pmrad [23].

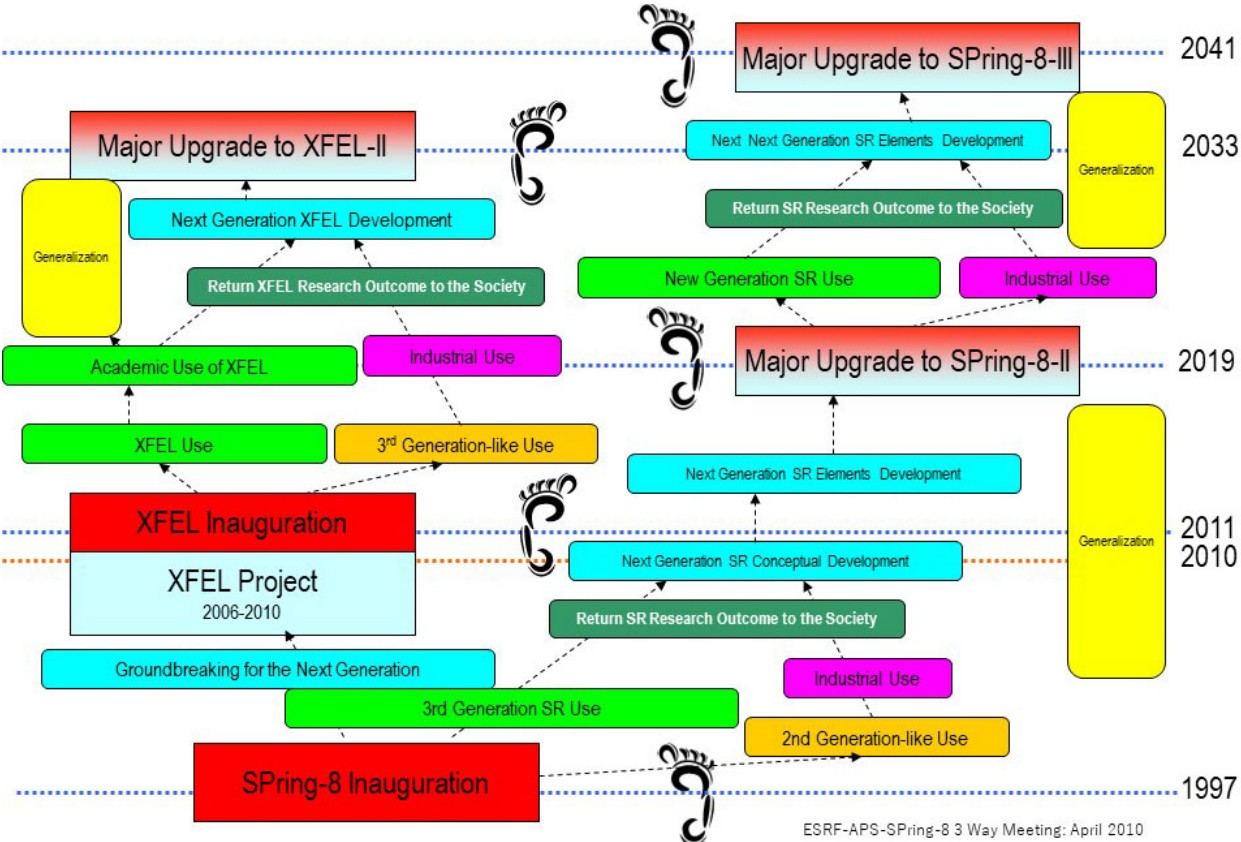

**Figure 7.** Roadmap for the SPring-8 site (as of April 2010). The SACLA was described as the XFEL Project. With the completion of SACLA, bipedal walking with SPring-8 became possible. In 2010, we planned to start upgrading SPring-8 in 2019, but this plan has been delayed.

However, after the Great East Japan Earthquake in 2011, constructing a synchrotron radiation facility as a symbol of the reconstruction in the disaster area became a hot topic. The decision to construct a 3 GeV synchrotron radiation facility in Sendai postponed the SPring-8- II refurbishment. The SPring-8/SACLA accelerator team has taken the lead in constructing the accelerator for the SR facility in Sendai, and work is underway, intending to start operation in 2024. The experience of constructing the Sendai facility has clarified

many points in terms of improving the SPring-8-II design. We hope to complete a highly stable synchrotron radiation facility with ultra-low emittance in the latter half of the 2020s.

SPring-8/SACLA issued the Green Facility Declaration in August 2021. It is parallel to the Japanese government's attempt to realize the declaration of achieving carbon neutrality in 2050 with the help of science and technology, and the government has stated that it will strongly support industry–government–academia activities toward carbon neutrality. In addition, we also announced that we would promote energy conservation at accelerator facilities that consume large amounts of energy. We have achieved significant energy savings by changing the SPring-8 from the previous booster synchrotron to the SACLA linear accelerator.

The fourth step following SPring-8-II is the upgrade of SACLA. This plan is still in the conceptual stage, but we are looking for a way to achieve a repetition rate of 10 kHz with a normal-conducting linear accelerator while maintaining the character of SACLA's compact XFEL. The application of dielectric acceleration developed by KEK would be one candidate. The fifth step would be SPring-8-III, but it is still unclear. Storage ring-based XFEL development is one candidate, but almost everything depends on future developments.

The author would like to thank all those who have contributed to making SACLA possible today.

**Funding:** This research received no external funding.

**Institutional Review Board Statement:** Ethical review and approval were waived for this study due to the historical nature of the article.

**Informed Consent Statement:** Not applicable.

**Conflicts of Interest:** The author declares no conflict of interest.

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
