# Peer review of "Early Days of SACLA XFEL"

_photonics, doi:10.3390/photonics9050357_

Round 1

Reviewer 1 Report

SACLA is the world's second built and operated hard X-ray free electron laser. 
SACLA plays a historically important role in the field of X-ray science. 
This manuscript covers the behind-the-scenes of design, initial construction and operations well, and I think it is very useful for understanding the history of XFEL and SACLA. In my opinion, the current form of the manuscript merits to be published in Photonics. 
The only thing I can suggest is to very carefully check the whole manuscript (before the final publication) in terms of English and any mistakes (see below)

minor
1. The red color of (Fig.X) should be changed to black text.
2. line 27: 'Stanford' should be 'Stanford Linear Accelerator Center (SLAC)'
3. line 122: 'medium[28].' to 'medium [28].'
4. Space between number and unit (e.g. line 231: '300m' to '300 m')

Author Response

Dear Reviewer 1,
Thank you for your valuable comments. I hereby address point-by-point your comments:

  1. The red color of (Fig.X) should be changed to black text. -> corrected.
  2. line 27: 'Stanford' should be 'Stanford Linear Accelerator Center (SLAC)'->corrected.
  3. line 122: 'medium[28].' to 'medium [28].'-> corrected.
  4. Space between number and unit (e.g. line 231: '300m' to '300 m') -> I have read through the manuscript and added all necessary spaces between numbers and units.
  5. In addition, I have added one sentence 'The author would like to thank all those who have contributed to making SACLA today.’ at the end of the manuscript.

Reviewer 2 Report

I read with great pleasure the paper by Ishikawa. The article describes, with a historical perspective, the process that lead to the construction of the SACLA FEL and its activities. In my opinion, this kind of papers are very useful to the scientific community, since they highlight strong and weak points of these large-scale experimental facilities.

In particular, this paper is very well written and it is for sure of interest for the readers of “Photonics”, therefore I can warmly recommend it for publication in the journal in its present form.

Author Response

Dear Reviewer 2,

Thank you for your valuable comments. I have minor revisions to the English language and style, according to the other reviewer. In addition, I have added one sentence, 'The author would like to thank all those who have contributed to making SACLA today.' at the end of the manuscript.